# Argumentative Reasoning: Development, Training, and Relevance to Academic Outcomes

**DOI:** 10.3390/bs15121700

**Published:** 2025-12-08

**Authors:** Robert Ricco

**Affiliations:** Department of Psychology, California State University, San Bernardino, CA 92407, USA; rricco@csusb.edu

**Keywords:** argumentative reasoning, deliberative argument, justification, counterargument, meta-level representation, epistemic beliefs, training effects, arguing to learn

## Abstract

Argumentative reasoning (AR) refers to the kind of reasoning used when individuals engage in argument about a disputed claim or proposed action. In its mature, most proficient form, AR involves several reasoning skills such as providing effective justification for one’s claims, anticipating and defending against challenges to those claims, and critiquing the position and reasoning of one’s opponent in the argument. Mature AR also involves an idealized understanding of argument as a rule-governed, rational process in which arguers seek to persuade one another through reasons, rather than through force or emotion. WE There is compelling evidence that proficiency in AR, resulting from natural development or from targeted educational experiences and training, is associated with better academic outcomes in middle childhood, adolescence, and emerging adulthood. These outcomes include greater depth and breadth of learning in specific content areas (e.g., science), more effective written communication, and higher order critical thinking. This article begins with a discussion of the nature and significance of AR, followed by an account of the link between AR and academic achievement in primary and secondary school. In the principal sections of the article, the development of AR skills is discussed along with the results of explicit efforts to train AR. The section on training effects includes a discussion of how motivation, culture, and gender influence student engagement in argument-based classroom activities.

## 1. Introduction

Within the fields of psychology and education there is a long-standing interest in the development of argumentative dialogue and writing ([76]; [109]). This interest stems, in part, from the recognition that classroom settings and pedagogical practices that support student engagement in deliberative argument foster deep, conceptual learning ([8]) and more advanced language skills ([21]). Engaging with subject matter in an argumentative manner and participating in argumentative dialogs with peers are effective ways to learn. But argument is more than just a means to an end. It is important for its own sake. In particular, argument plays a central role in science ([97]) and in civic engagement within a democratic society.

Researchers within each scientific discipline engage in substantial debate about whether a given hypothesis or theory has been empirically supported over and above alternative explanations. Such debate takes the form of deliberative, rational argument and it is the basis for a researcher’s critique of prior research in the field and for their response to anticipated or real critiques of their own findings. The real test of knowledge claims in science lies as much in the process of formal and informal argument among peers as it does in the researcher’s strict adherence to the scientific method ([23]; [22]). AR is no less central to effective participation in a democratic society. Both participatory and representative democracies depend, crucially, on the free practice of deliberative argument within the various public spheres of society.

Argument has additional significance beyond its role in science and civics. The unique form of reasoning used in argument—argumentative reasoning (AR)—lies at the heart of most accounts of critical thinking. Recent calls for a greater emphasis on the teaching of critical thinking skills in primary and secondary school are at one and the same time calls for training in the use of AR ([69]). Proficiency in judging the accuracy of information gleaned from various media, and proficiency in determining whether a claim is sufficiently supported by the available evidence, are both important goals of a critical thinking curriculum ([17]) and they both involve AR. Ensuring that students develop competence in AR has a particular urgency to it arising from the fact that discourse in our highly politicized and polarized public spheres often blurs the distinction between opinion and fact and between belief and truth ([35]). Making competence in AR a general goal of education is the best way to reestablish these distinctions and to uphold them.

This article provides a review of the literature relevant to the following questions: (a) Is proficiency in AR associated with better academic outcomes in primary and secondary school? (b) How does AR develop and what competences are present in childhood and adolescence? (c) Can AR be trained and, if so, which methods of training are most effective? (d) How do motivation, culture, and gender influence the extent to which students engage in, and benefit from, training in AR?

There are several extant reviews of the training methods or pedagogical practices employed to assist students in learning to argue ([59]; [109]; [132]) and arguing to learn ([8]; [133]). While training methods and their effects will also be discussed here, the primary focus of the current review is on the development of AR and on how the findings from training studies shed light on its development.

### Method

Multiple databases were searched in preparing this review including PsycINFO, ERIC, and Science Direct. Search terms varied depending on the specific section of the review and are indicated in Table 1. Papers resulting from these searches were then required to meet the following criteria: (1) Publication between 1995 and 2025 for papers reporting empirical research. (2) Inclusion of data from primary and/or secondary students for papers reporting empirical research. For the section on the development of AR, papers featuring research with children of preschool age were also included. (3) Publication in academic or professional journals or in books by an academic publisher. (4) Inclusion of a control group for papers reporting the effects of training.

## 2. What Is Argumentative Reasoning (AR)?

AR refers to the kind of reasoning used when people engage in argument about a disputed claim or proposed action. In its mature, most proficient form, AR involves several reasoning skills such as providing effective justification for one’s claims, anticipating and defending against challenges to those claims, and critiquing the position and reasoning of one’s opponent in the argument ([74]; [117]).

Table 2 provides brief definitions of key elements of AR and illustrates several of these with an argument fragment adopted from an online discussion thread. In Turn 1 of this dialogue, Andrea makes a *claim* that meat is unhealthy, and she provides as *justification* the statement that meat carries multiple cardiovascular risks. Brenda advances a *counterargument* to the effect that excessive meat may be unhealthy, but as part of a balanced diet meat carries no risks. In her *rebuttal* to Brenda’s counterargument, opening Turn 2, Andrea chooses not to defend her statement from Turn 1 that eating meat leads to obesity, high blood pressure, etc. Instead, she opens a new line of reasoning focusing on the supposed chemicals in meat. Finally, Brenda closes out this argument fragment by posing a counterargument to Andrea’s statement that meats contain chemicals.

The dialogical nature of AR is its most significant feature and the one that distinguishes it from several other forms of reasoning ([12]; [136]; [94]; [117]). Deductive reasoning, for example, is generally construed as a monological process involving a working out of the implications of a single position represented by the premises taken as a totality. The participants in an argument, however, do not pursue separate, independent lines of reasoning. They seek to convince an opponent who may be attempting to support their own counter-thesis and who is free to raise objections. Each of the two opposing lines of reasoning in an argument develops in response to the other ([92]; [112], [113]; [120]).

In addition to specific reasoning skills, mature AR also involves an idealized understanding of argument as a rational, rule-governed, and truth-oriented process involving persuasion through reasons, rather than force or emotion ([55]; [80]). This understanding includes commitment to a procedure that allows claims and reasons to be freely given and mutually critiqued ([135]; [142]). Arguments possessing these characteristics are commonly referred to as *deliberative arguments* and can be contrasted with eristic arguments where the goal is simply to ‘win’ by whatever means prove effective ([3]). Eristic arguments are not bound by mutually shared norms of rationality or truth. In practice, deliberative argument can only be approximated to varying degrees. Real arguments always carry the potential to devolve into a contest of wills.

There is an adversarial aspect to deliberative argument, but this obtains within a broader collaborative framework driven by the common purpose of arriving at the truth of the matter. In deliberative argument, participants pursue twin goals. They seek to support and defend their own position in the dispute (goal 1) while questioning and critiquing one or more opposing or alternative positions (goal 2) ([79]; [142]). Both of these activities are essential to determining the merits of each participant’s argument and to any possibility of reaching a consensus on the disputed matter. Consensus, in and of itself, however, is not the goal of deliberative argument since consensus can be reached prematurely and in the absence of sound reasoning and true claims.

## 3. Arguing to Learn: Academic Outcomes of AR

Several aspects of AR are widely considered to be essential features of critical thinking ([42]; [69]). These include constructing and critiquing arguments, establishing whether a claim is sufficiently supported by the available evidence, distinguishing between sound and fallacious inferences ([17]; [95]), and recognizing elements of bias in one’s own reasoning and in the reasoning of others ([17]; [149]). Accordingly, the development of AR is, at the same time, the development of core components of critical thinking.

### 3.1. Writing

In addition to promoting critical thinking, AR has been linked to advances in writing and conceptual learning. Dialogue-based interventions such as the Argue With Me (AWM) paradigm have been found to improve the writing of students in middle school and high school particularly with regard to the essay and science writing genres (e.g., [21]; [71]; [80]; [76]; see also [74]). Though both of these types of writing are inherently dialogical and stress the consideration of alternative positions on an issue, there is a strong tendency for students to limit their focus to a single, favored position and to neglect alternative or opposing positions ([34]). This is especially true in primary school ([129]). Students also tend to neglect potential problems or inconsistencies within their adopted position. The medium of writing does not typically feature prompts to consider both sides or to address limitations within one’s favored position. By contrast, the activity of dialogue prompts, to varying degrees, the consideration of alternative positions and perspectives and an assessment of their strengths and weaknesses. It does so by way of challenges, requests for clarification, assertions of counter claims, etc. The extent to which this is the case varies with the characteristics of a given dialogue. Sustained, intensive experience with argumentative dialogue combined with reflective activities that promote a meta-level understanding of deliberative argument support the internalization of dialogic prompts making them available to the act of writing ([44]). This results in discernable improvements in students’ essays including greater articulation of positions that differ from that of the writer, more use of counterargument, more effective defenses against potential challenges, and more sustained efforts to integrate opposing positions ([21]; [58]; [74]; [73]; [105]; [122]).

### 3.2. Content Knowledge

Why might the development of AR support conceptual learning within a given content area or domain? There appear to be multiple reasons. First, dialogic argument and argumentative writing are forms of communication and require the individual to reflect upon and objectify concepts in order to articulate them with precision ([26]). Although true of communication in general, this element of reflection has been found to make the speaker aware of limitations in their understanding of a concept. Second, the need to justify and defend one’s claims, as required in argument, can help to revise and improve those claims ([26]; [20]; [140]) and can motivate a search for new information. Third, exposure to alternative claims and their supporting reasons introduces new concepts as well as new interpretations of familiar evidence ([20]). Finally, efforts to integrate seemingly opposed positions, should arguers seek consensus, involves deep processing of information and the formation of new connections among concepts.

Evidence of gains in content knowledge within a specific domain resulting from participation in dialogic argument training or instruction in the elements of AR is common. Such interventions (relative to controls) have been found to promote a better conceptual understanding in domains such as physical science ([6]; [140]; [49]; [56]; [87]; [145]; [153]; [156]), social science ([81]), mathematics ([63]), history ([150]; [151]), and literature, ([61]) as well as selected debate topics (e.g., [58]). In some cases where learning gains following AR training were not found on an initial post-test, delayed gains were present on a second post-test (e.g., [7]; [46]; [81]).

There is also evidence of learning gains being associated with the use of specific AR skills. For example, the use of counterargument, and the adequacy of students’ attempts to justify a claim predict gains in learning ([20]; [82]). Similarly, students showing greater gains in conceptual learning are more likely to use reasoning that acknowledges and responds to the contributions of the dialogue partner, especially challenges, defenses, requests, and agreements ([7]).

Learning gains are by no means guaranteed, however, and appear to be dependent upon several factors. Some research studies have found gains in AR skills following training, but no learning gains in the targeted subject matter ([144]). [8] ([8]) point out that a number of studies failing to find learning gains focus on superficial learning outcomes (knowledge of facts) and not outcomes such as conceptual learning that require deeper processing or the correction of misconceptions. In addition, the content domains for some of these studies are not especially complex.

## 4. Development of AR

In order to track the development of AR, it is important to distinguish between those skills that are displayed by the child independently, without guidance or training, and those that are only present in the child’s performance when some manner of support is provided. From a developmental standpoint, we can speak of three broad phases in the acquisition of a skill. In the first phase, the skill is not available to the child even with support. In the next phase, the skill is displayed when the child is provided with a particular form of support and not otherwise. Finally, in the last phase, the child performs the skill independently, i.e., without the need of support. A skill or type of knowledge that is displayed spontaneously or independently can be described as fully consolidated ([84]) or as fully internalized ([141]).

While purely observational studies of AR among age groups in primary and secondary school are common, the research literature on AR is also replete with attempts to train AR skills and knowledge. Such research typically assigns children of a certain age to either a control group or an experimental group, where the latter receives training and the former does not. The assignment may be random or non-random. The groups are compared on one or more measures of AR skills or knowledge in order to determine the effectiveness of the training. How should we apply the findings of such training studies to the question of how and when various AR skills develop? AR skills or knowledge present in the performance of a control group, i.e., a group that receives no training, or that are present in performance on a pre-test, i.e., prior to the introduction of a training intervention, can be considered fully internalized or consolidated for the age group being assessed. By contrast, skills that are only displayed by children who have received training, i.e., children in the experimental group, can be considered skills that children in this age group are capable of acquiring or applying with the help of targeted training. The extensiveness of the training needed to demonstrate the skill and the extent to which the skill can be transferred to conditions that differ from those involved in the training program indicate the extent of consolidation or internalization at that age.

### 4.1. Evidence of the Independent Use of AR Skills

In what age range do we typically see a particular skill being displayed independently? The research summarized here allows the assessment of the independent use of AR skills by the child or adolescent. These findings stem from the observation of children in natural settings or performing contrived tasks in research settings. They also include the assessment of children in a control condition or pretest as part of a training study.

#### 4.1.1. Justification

One prominent theory of argument ([88]; [89]) maintains that reasoning for the human species serves the purpose of persuading and assuring others, thereby establishing the state of consensus needed for collective social action. In this view there is an evolutionary basis to AR and we should see early evidence of reasoning skills in dialogic settings. Indeed, young children exhibit several AR skills in basic form. They spontaneously offer justifications for their claims, and they can defend a claim or its justification if challenged. Multiple studies provide evidence that the competence to justify claims is present in family conversations or play interactions in early childhood ([2]; [43]; [125]). Although preschool children’s dialogues often leave certain elements of argument unstated, these can be readily reconstructed to yield coherent arguments ([1]). Early use of justification is relatively common in settings that are familiar to children or that have personal relevance to the child ([2]; [29]).

Several competencies important to providing effective justifications for claims are present in early childhood. These include a basic sensitivity to *circular reasoning*, *common ground*, and *source reliability*. Young children are sensitive to whether a justification adds anything above and beyond a simple restatement of the claim. Justifications that largely repeat the claim are referred to as circular or ‘begging the question’. Five-to-six year olds show a preference for noncircular over circular arguments, indicating some appreciation of what makes for better reasoning in argument ([11]; [19]).

Five-year-olds are sensitive to *common ground* in conversations about physical objects. They are able to recognize whether or not they share, or have established, common ground with their conversational partner and they are more likely to make explicit the connection between a claim and its supporting evidence when common ground cannot be assumed. For example, when building a zoo and explaining why they placed a polar bear in an exhibit featuring ice, five-year-olds indicated that this was because “it’s ice” or it’s a polar bear,” leaving out the warrant that explains the connection (“polar bears need ice”). This suggests that they knew the warrant was common knowledge. However, when explaining their preferred location for objects that are not conventional for zoos (e.g., a washing machine placed with the lions and tigers), they provided an explicit reason (“so the animals could wash their coats”) ([67], [65]).

Young children can assess the *reliability* of an information source. [68] ([68]) found that five-year-olds but not three-year-olds distinguished between a person who was an eyewitness to an event and a person whose knowledge about the event was second hand. The five-year-olds relied on the former source but not the latter, and they also talked explicitly about the reliability issue with peers. Seven-year-olds and less so five-year-olds have been shown to pick partners to collaborate with whom they know to have preferred reliable (eyewitness) over unreliable (second-hand) testimony. This was the case even though the partner’s choice of the more reliable source did not help them in the end ([27]). Similarly, children as young as four years can distinguish between a source who can verify her claim about the location of a toy (e.g., because she looked in all available containers) and one who can only partly verify her claim (e.g., she looked in some but not all the containers) and they give greater credence to what is reported by the former person. In addition, children who do so also tend to provide explicit explanations of the reliability issue involved ([16], [15]).

#### 4.1.2. Advancements in Justification

Though a precocious AR skill, evident in early childhood, the use of justification develops through middle childhood and adolescence, becoming more complex and more effective. One difference in justifications between younger (3 to 5 years) and older (6 to 9 years) children is with regard to whether the justification is “activity-bound” or “activity-unbound”. The former makes reference, solely, to the child herself while the latter makes reference to something or someone other than the child such as a third party, norm, or rule. Activity-unbound justifications are more typical of the primary school years than of early childhood ([2]).

Justifications that involve more than one reason for accepting a particular claim are not common in student writing during middle childhood, and even when more frequent, generally in early adolescence, justifications using more than one reason tend to consist of unrelated reasons, i.e., reasons that provide fully independent support for a claim. Only later do we see frequent use of justifications involving reasons that are related or coordinate in their support for the claim ([25]). In addition, children are not always aware of, or able to explain, how the evidence they state justifies the claim it is meant to support. Even 13- to 14 year olds have been found to struggle in this respect ([40]). It is also the case that skill in justifying claims in a students’ own essay writing does not guarantee success in identifying the main claim and supporting evidence in a text ([85]). Thus, students’ use of justification is far from mature in the primary years.

#### 4.1.3. Counterargument

Somewhat more controversial than evidence of justification in early childhood is whether young children also use counterargument. Consider the following research paradigm, typical for studies reporting the early presence of counterargument. Pairs of preschool children were told about a girl who must walk to school in the rain. The children were then asked which of two boxes contains what the girl needs. One box contained an umbrella while the other contained boots. One child in the pair was surreptitiously told that the umbrella in the box was broken and that the boots had polka dots. When their uninformed partner suggested opening the umbrella box because the umbrella would be most helpful, informed five-year-olds (but not three-year-olds) countered that the umbrella was broken. By contrast, when their partner suggested opening the boot box, the five-year-olds did not use the fact that the boots had polka dots in a counterargument ([66]; See also [108]), suggesting that children of this age can counterargue and can distinguish between good and bad bases for counterargument.

Despite the presence of a basic form of counterargument in early childhood, and evidence that counterargument is trainable in the late childhood years ([62]), the spontaneous use of counterarguments aimed at critiquing an opponent’s position is not common in research or educational settings prior to early-to-mid-adolescence ([55]; [64]; [74]; [122]). Pre-adolescents show a lack of awareness of alternatives to their own claims and have difficulty critiquing their claims so as to anticipate potential challenges. Thus, for example, there is little recognition that acknowledging a potential counterargument and rebutting it effectively would increase the persuasiveness of one’s argument. When engaging in argumentative dialogue or when writing essays, elementary and middle school children focus primarily on justifying their own claims and defending those claims against challenges made either by their partner in the dialogue or by a third party ([34]; [138]).

How can we reconcile evidence of the deliberate use of counterargument in early childhood with evidence of its relative infrequency in middle childhood? Although a definitive answer to this question awaits further research, there are several viable explanations. First, the examples of young children employing counterargument in research settings involve non-complex situations featuring basic properties of concrete objects. By contrast, evidence for the relative absence of counterargument in middle childhood generally involves more complex situations such as determining the solution to a problem in physical science or resolving a social or moral controversy. The topics and concepts central to the use of counterargument in these complex situations tend to be abstract and sometimes limited to formal educational settings. All of this makes the use of counterargument more demanding relative to the conditions under which we find evidence of this AR skill in early childhood ([75]). Second, the more formal use of language typical of classroom settings is still relatively new to children in the primary grades. It is difficult for children to display a reasoning competence such as counterargument in a formal language that they are still in the process of mastering and that differs from the language they use in natural, non-school settings. Third, developing proficiency at articulating and justifying a favored position, a skill that develops during middle childhood, does not necessarily entail an equal ability to appreciate and critique alternative or opposing positions ([104]; [124]). In order to consistently employ counterargument, one must appreciate that challenging opposing claims is just as important as supporting one’s own claims and that justification and counterargument are closely related elements of AR. Such an appreciation requires a relatively explicit understanding of deliberative argument, something which is rudimentary at best in middle childhood ([55]; [80]).

As has been evident since [51]’s ([51]) seminal research on adolescence, this age period represents a time of substantial cognitive development. This includes important advances in reflection and deductive reasoning, each of which contributes to the development of argumentative reasoning. Regarding reflection, a key, gradual development in adolescence involves the consolidation of a meta-level understanding of deliberative argument. This includes an awareness of the twin goals of argument, i.e., the importance not merely of supporting one’s own position but also of considering and critiquing any opposing or alternative positions ([79]; [142]). This meta-level understanding makes possible the intentional constraint of reasoning by the twin goals and the strategic, coordinated use of AR skills in service of those goals. Thus, it is in adolescence that we see more spontaneous, unprompted consideration of the claims being asserted by one’s dialogue partner in a disagreement or dispute and more efforts at critiquing those claims ([21]). Such efforts include the anticipation of potential objections to one’s position and a greater reliance on the use of counterargument.

The emergence of a formal understanding of deductive or logical validity in adolescence also contributes to this newfound reliance on counterargument ([115]). Adolescents are more likely than younger individuals to recognize that supporting a statement (P) by showing its negation (not-P) to be false is a valid strategy in deductive reasoning while supporting a statement by providing evidence consistent with the truth of the statement is not ([38]). For the most part, deliberative arguments are not deductively valid, and they needn’t be in or to be convincing. However, the use of counterargument in informal reasoning is analogous to the use of a falsification strategy in formal, deductive reasoning. For this reason, an understanding of the merits of falsification contributes to an appreciation of the importance of counterargument.

In addition to an explicit understanding of the twin goals of deliberative argument, and of the role of counterargument in pursuing those goals, adolescents also develop an appreciation of the procedural rules regulating deliberative argument. These rules, when followed, ensure that deliberative argument does not devolve into eristic argument ([137]; [143]). Examples of such rules include 1. claims must be supported through the provision of reasons rather than the use of power-based tactics or emotional appeals, and 2. anyone advancing a standpoint has the obligation to defend it if requested to do so ([134]).

Informal reasoning fallacies such as ad hominem, ad ignorantiam, ad populum and straw person arguments represent violations of these procedural rules. Ad hominem arguments attack the arguer rather than the arguer’s claims or reasoning. This violates rule (1) prohibiting the use of power-based tactics. Ad populum arguments claim that the popularity of a position is a sound basis for accepting it while ad ignorantiam arguments maintain that if we don’t know that a claim is false then we can conclude that it is true. Each of these latter two fallacies violates procedural rule (2) requiring that advancing a claim carries a burden of proof which must be met to the satisfaction of one’s audience. Knowledge of these procedural rules and specific reference to them in identifying and rejecting ad ignorantiam, ad hominem, ad populum arguments increases from seventh through eleventh grade ([148], [147]). These findings show how a developing understanding of deliberative argument becomes part of students’ ability to distinguish between valid and fallacious reasoning.

#### 4.1.4. Epistemic Beliefs

Epistemology is a branch of philosophy that studies the origins and nature of knowledge. Adolescence brings the emergence of more sophisticated beliefs concerning knowledge and knowing and this advance contributes to the development of AR ([80]; [54]; [155]). There is extensive research showing that more proficient and consistent performance on argumentation tasks is associated with more advanced epistemic beliefs ([70]; [114]; [146]). One key advance in this regard is coming to understand that valid knowledge is the result of an objective and unbiased justification procedure. This understanding, in turn, gives importance to the practice of subjecting one’s beliefs and those of others to scrutiny and rejecting or modifying beliefs accordingly ([119]). A second advance concerns the abandonment of an absolutist, black-and-white view of argument in which one side is right and the other wrong and the development of an evaluativist view in which both sides have legitimacy, but a defeasible judgment in favor of one side is typically possible on the basis of available knowledge ([70]; [72]; [86]).

Progress in understanding the nature of knowledge has profound implications for students’ motivation to participate in deliberative argument. If students espouse an absolutist conception of argument wherein one side is deemed to ‘fit the facts’ while the other side merely has there facts wrong, then there is little purpose to engaging in argument. A process of fact-checking would be sufficient. Likewise, until valid knowledge is understood as the result of a standardized and transparent justification procedure, there is little point to argument since there is no basis for deciding between competing claims.

### 4.2. Working Memory and Cognitive Control

Engaging effectively in deliberative argument would seem to place substantial demands on students’ ability to retain and manipulate information in working memory. Likewise, this form of dialogue typically requires multiple cognitive control processes including planning (e.g., strategizing), updating the contents of working memory, inhibiting irrelevant information, and shifting between perspectives ([37]). Working memory capacity and cognitive control are both subject to development ([39]; [118]). This raises an important question. To what extent does the development of working memory and cognitive control contribute to the development of AR? Could the relative lack of consideration of opposing positions found in the arguments of primary school children be due to the information processing limitations of this age group? Surprisingly, there is very little research that addresses this question.

[104] ([104]) developed a theoretical hierarchy of AR skills based on the presumed demands each skill places on working memory capacity. These demands constitute the cognitive load of a particular skill. Skills that require the coordination of two or more elements involve higher cognitive load than skills involving only a single element. On this basis, they argued that constructing (or critiquing) a justification involves the smallest cognitive load. Articulating the link between a claim and its justification, i.e., explaining why the stated reason or evidence supports a particular claim, involves moderate cognitive load. Constructing a counterargument, or rebutting a counterargument, involves still greater cognitive load. [104] ([104]) then provide evidence that the development of AR skills follows this progression. The more elements that must be coordinated in executing a skill, the later this skill is acquired. However, the hypothesized cognitive load of the various AR skills has not been verified through testing. Thus, it is not clear that the progression reflects increasing demands on working memory per se.

[79] ([79]) provide indirect evidence that developmental constraints on working memory or cognitive control cannot fully explain age differences in addressing opposing perspectives and engaging in counterargument. In an effort to reduce the cognitive load posed by the spontaneous use of AR skills in an ongoing dialogue, Kuhn and Udell assessed the AR reasoning of seventh and eighth graders without requiring them to engage in dialogue. Students had to choose the better of two arguments where one argument supported position A while the other argument refuted an opposing position B. Thus, the choice was between supporting a position directly (justification) or supporting it indirectly by challenging the opposing position (counterargument). Both age groups only infrequently chose challenging their opponent as a strategic option and did not generate many counterarguments of their own when asked how best to support a given position. Thus, the relative lack of counterargument in the dialogues of young adolescents remains present even when the information processing demands of the dialogue or discourse setting are reduced. The authors argue that these findings support the notion that the lack of counterargument and other reasoning that compares and contrasts opposing positions in early adolescence cannot be due, exclusively, to developmental constraints on information processing capacity. The authors suggest that the primary factor explaining the absence of counterargument is a lack of awareness of the need to use counterargument—a lack in metacognitive awareness of the twin goals of deliberative argument.

### 4.3. Summary

The use of reasons or evidence to justify a claim or to defend it when challenged is present in early childhood, as are three competencies that are important to the effective use of justification. (1) Young children understand the difference between a claim and its justification, and they recognize that the latter must do more than merely restate the claim. (2) Also, they can judge what knowledge they hold in common with a conversational partner and, therefore, whether the link between their claim and its justification needs to be explicitly stated. (3) Finally, they consider the reliability of a source in deciding what weight to give to that source’s testimony.

Though available in early childhood, the use of justification continues to develop through adolescence. Advancements in this AR skill include the emergence in middle childhood of an appeal to third parties, rules, or norms in justifying a claim. Additional refinements in adolescence include the use of multiple reasons and an appreciation of whether those reasons are independent of one another in their support for the claim or whether they are interdependent in their support.

The occasional and rudimentary use of counterargument is arguably present by the age of five or six. Multiple research studies report that counterarguments can be readily elicited in children of this age and that six-year-olds distinguish between sufficient and insufficient bases for making a counterargument. Somewhat paradoxically, however, the spontaneous use of counterargument is not common prior to mid-adolescence. Perhaps the primary reason for this is that the strategic use of counterargument requires an appreciation of how central this AR skill is to the goals of deliberative argument. Critiquing opposing positions is just as important as supporting one’s favored position in an argument. This meta-level understanding of deliberative argument is late developing and consolidates sometime during middle to late adolescence. An appreciation of the procedural rules of deliberative argument is also associated with adolescence and supports the identification of informal reasoning fallacies.

The extent to which the development of working memory and cognitive control contribute to the development of AR is relatively unstudied. Consequently, it is unclear whether the relative absence of counterargument in the primary grades is due to age-based limitations in working memory and cognitive control. The lack of research into this question, notwithstanding, however, use of computer-based scaffolding to train AR in early adolescence aims to compensate for processing limitations in this age group. This type of training will be discussed in the next section.

## 5. Training Effects

Having some sense of when various aspects of AR are displayed independently or spontaneously by children and adolescents, we turn now to the question of whether AR skills can be trained and at what age training is successful. If children of a given age do not display a particular skill independently but do use it successfully following training this suggests that there is a readiness in that age group to acquire the competence. Alternatively, this could indicate that the competence is present but constrained by contextual factors.

### 5.1. Training Effects with Middle Schoolers

The largest body of data regarding attempts to train AR derives from the Argue With Me (AWM) paradigm ([59]). In its most common format, this training program consists of three phases. In a pre-game phase students work in pairs with a same-minded peer to identify reasons for their position on a topic and evaluate the strength of these reasons. They also consider possible opposing positions and anticipate viable critiques of their position. The game phase involves several sessions of online argumentation with an opposing pair of students. Finally, the end game phase involves extensive preparation for participating in a full class discussion of the topic. That preparation includes an opportunity for students to review and reflect on the contributions of their own pair and of the opposing pair to the dialogue of the game phase. Following the preparation sessions, the student dyads participate in a final debate with dyads from the opposing side. Each member of the pair debates, individually, with one member of the other pair. Finally, students write an essay arguing for their position on the topic. The game phase and end game phase also include reflective exercises where participants reviewed the arguments that had taken place.

Across several papers, Kuhn and colleagues report the results of a 3-year training study with children who were 11–12 years at the outset and 14–15 at the conclusion ([21]; [71]; [80]; [76]; see also [74]). A version of the AWM paradigm was used. Comparisons between the training and control groups indicated similar training effects in both written essays and dialogues, though the timing of these effects varies across the two media.

Regarding *essay writing*, training led to 1. a greater tendency to critique opposing positions and to bring both the participant’s position and the opposing position into relation with one another, and 2. greater appreciation of the importance of evidence in weighing the merits of one position relative to another. With regard to *dialogue*, training effects included 1. greater use of a counterargument strategy involving direct critique of claims made by the opponent, and 2. more frequent use of meta-talk—commenting about the ongoing argument. An increase in the amount of meta-talk occurred in the training or intervention group across the first year and these gains were generally maintained in the second year. Such meta-talk included directive statements (“You need evidence…”), evaluative statements (“You are only asking questions…”), and statements regarding comprehension (“Your statement makes no sense.”). The developments concerning meta-talk suggest increasing awareness of, and commitment to, the goals and norms of deliberative argumentation across the course of the training. There was little evidence of gains in each of these respects in the control group. Finally, the training group also showed better appreciation of the importance of evidence and the desirability of having multiple sources of evidence where possible.

Similar effects of AWM-based dialogic training with middle schoolers have been found with briefer interventions. Across the course of a year, [73] ([73]) found improvements (relative to controls) with regard to the spontaneous use of counterargument and rebuttal in both the medium of electronic dialogue and written essays. Meta-level gains in electronic dialogue were also evident including better appreciation of the goals of argument. Following 16 sessions of dialogic training with eighth graders across 12 weeks, [78] ([78]) report that the training group was less likely than the control group to use justification of one’s own position as their sole argumentation strategy. The training group was also more likely to criticize the opponent’s statements and to offer alternatives to the opponent’s claims. Dialogic interventions as brief as five weeks ([33]) have resulted in post-training dialogues that feature a greater “range” of arguments, more two-sided arguments, greater awareness of the possibility and legitimacy of multiple views on an issue ([77]), and increased use of counterargument and rebuttal ([33]; [52], [53]).

Generally absent or less clearly present in training studies with young adolescents were gains in skills and strategies more typical of adult argumentation such as the use of leading questions to gain commitment to claims consistent with one’s position ([33]), blocking or undermining the leading questions of one’s opponent ([33]; [78]), and directing the opponent’s reasoning in order to have them make statements that might reveal an inconsistency ([78]).

Are the effects of training transferable? Evidence that training effects within a particular medium (e.g., face-to-face dialogue) or content area (e.g., social science) transfer to a different medium (e.g., writing) or content area (e.g., physical science) would strongly suggest that the skill displayed following training has been consolidated or internalized. Transfer of AR skills from computer-mediated or face-to-face dialogues to essay writing has been well established by the research reviewed above. But is there evidence that the AR skills facilitated by the AWM training in a particular domain transfer to a different domain? As discussed by [59] ([59]), transfer of counterargument and other AR skills following training has been found from one physical science topic to another ([57]), one social science topic to another ([73]) and from a social domain to a science domain and vice versa ([52]).

### 5.2. Training Effects in Middle Childhood

Evidence of successful dialogic training of counterargument in fourth and fifth grades is also available, though, overall, the findings are somewhat mixed. Using dialogic training based in the Collaborative Reasoning (CR) approach, [62] ([62]) found effects, relative to controls, on fourth graders’ written essays. These effects included more counterarguments and rebuttals along with fewer irrelevant statements. Similar findings with essay writing are reported across several other studies presenting children in these grades with from four ([28]; [62]; [110]) to ten ([111]) CR training sessions. Consistent with these results, [90] ([90]) found that giving fifth graders a goal of communicating with an audience in revising their texts made it significantly more likely that they took the opposing position into account and sought to rebut it.

By contrast with these studies, [10] ([10]) found effects of dialogic training with 11-year-olds that were limited to the use of justification. The benefits of training did not extend to the use of counterargument. And in a study featuring mixed results, the use of prompting enabled fourth graders to produced more alternative standpoints and more rebuttals of these, but the students’ arguments were described as “relatively shallow and poorly developed”. In addition, the students did not generate counterarguments to their own standpoint ([36]).

### 5.3. Computer-Assisted Scaffolding

As noted previously, the precise role of working memory in the acquisition and use of AR skills remains an open question. Nonetheless, the design of computer-based scaffolding techniques as a means of training AR skills in classroom settings has been based, in part, on the assumption that reducing cognitive load can compensate for age-based (developmental) or individual limitations in working memory capacity. These techniques have proven especially effective with students in sixth through eighth grades. In particular, providing paired down, straightforward sentence openers or prompts for eliciting key components of argument such as claims (“I think…” or “I believe…”) and justifications (“My reason is”) has enabled students in these grades to engage in deliberative arguments with peers at a more complex level than was otherwise evident ([47]; [83]; [152]; [153]). The use of concept maps and diagrams to represent an argument’s structure and the use of scripts, templates and frames to depict how each turn in the dialogue responds to another has also been effective in reducing cognitive load ([49]). Finally, the presentation of the same elements of argument through each of several media such as text, video, and illustrations is an effective way to provide redundancy while also taking into account the learning preferences of students ([83]). The fact that evidence for the effectiveness of computer-based scaffolding is more common in middle school or early adolescence than in the primary grades suggests once again that there is a developmental readiness during the former grades to benefit from training.

### 5.4. Conclusions Regarding Training Effects

The results of attempts to train AR indicate that the most consistent success in training counterargument and a broader consideration of the merits of alternative positions has been found in early adolescence. Some success is evident with fourth and fifth graders as well, but the findings are fewer and less consistent. The middle school years—early adolescence—is a time when students are ready to benefit from training. The skills are on the verge of consolidation. Transfer effects across media and content area are most commonly found in this age period which also implies that children of this age are capable of acquiring these skills in a form that is generalizable and self-regulated.

The interdependence between the use of counterargument and other moves that address alternative positions, and a meta-level understanding of deliberative argument is evident from findings that the training of AR skills leads to a greater awareness of the norms ([52]; [122]) and goals ([80]; [103]) of deliberative argument. Likewise, efforts to develop students’ meta-level understanding of argument through direct instruction or by having students reflect on the contents of their online dialogues with fellow students lead to improvements in AR skills above and beyond the effects of training in dialogic activities alone ([33]; [55]; [122]; [60]). In this way, the effective and strategic use of reasoning skills and a relatively explicit understanding of deliberative argument are two sides of a coin.

## 6. Student Characteristics That Influence the Training of AR

The extent to which students engage with, and benefit from, classroom activities designed to help them acquire AR skills is affected by a number of factors. An exhaustive review is beyond the scope of this paper. Instead, the focus will be on three student characteristics. These are student motivation, cultural values, and gender.

### 6.1. Motivation

Some students may be dissuaded from participating in face-to-face dialogical argument because of their academic achievement goals. Achievement goals describe a general orientation in approaching academic tasks and include performance goals and learning or mastery goals ([30]; [31]). Individuals who pursue mastery goals view academic tasks as opportunities to increase their competence, ability, or knowledge in a domain whereas individuals with a performance goal orientation interpret tasks as opportunities to receive feedback concerning their competence or ability level ([30]; [91]). Evidence that performance goals are sometimes associated with high achievement ([31]; [32]) has led goal theorists to distinguish between performance-approach goals and performance-avoidance goals ([31]). Approach goals involve an attempt to maximize positive feedback concerning one’s competence while avoidance goals involve an effort to minimize negative feedback. While approach goals might support learning, under certain conditions, both goal types make learning secondary or even incidental to performance.

[24] ([24]) and [4] ([4]) provide evidence of an association between a performance goal orientation and a focus on the interpersonal elements of argument. That is, students with performance goals emphasize the social setting of the argument and the relationship between the argument’s participants rather than the positions of the arguers. Students with performance-approach goals tend to interpret arguments as competitions and emphasize the adversarial elements of deliberative argument. Performance-avoidant goals are associated with an inclination to reach consensus prematurely, i.e., to agree with one’s opponent despite not being convinced by the evidence. By contrast, a mastery goal orientation is associated with an understanding of argument as a means for establishing the legitimacy of multiple perspectives and for arriving at valid knowledge. Students with mastery goals are more likely to approach class discussions from the standpoint of deliberative argument ([4]).

### 6.2. Culture

Cultures differ in how they view argument and these differences influence how students respond to argument-based classroom activities. One common dimension along which cultures are compared is individualism-collectivism ([130]). Collectivist cultures, typified by the countries of East Asia, prioritize the needs of the group over those of the individual and use a style of communication that is relatively indirect and reliant on context and shared knowledge ([41]; [127]). By contrast, individualist cultures, typified by the United States and Northern European countries, give priority to the individual over the group and favor a communication style that is direct and minimally dependent on context or shared knowledge. The fact that members of collectivist cultures can rely on context and can presume that they share considerable knowledge with their conversation partner reduces the need for argument. Consequently, members of collectivist cultures may view argument as disruptive of social bonds, and as an indication that normal channels of communication have broken down ([127]). This may explain findings that East Asians perceive themselves to be less argumentative than Westerners ([48]). These findings suggest that students from more collectivist cultures may sometimes need to be convinced that deliberative argument is a worthwhile and productive activity.

Differences in communication styles also underlie East–West differences in the degree of elaboration used in justifying a claim. Western students are sometimes found to be less succinct and more elaborate in their justifications than East Asian students ([126], [127]; but see [128], for a null finding). This may be because Westerners cannot assume that they share substantial knowledge with their dialogue partner and they cannot depend on their partner’s use of context to understand their intended meaning.

Another distinction that maps onto the East–West contrast is that between inductive reasoning based in judgments of plausibility or likelihood and deductive reasoning based in formal, logical relations among statements. East Asians have been found to prefer a more inductive approach to problem-solving in situations that do not require the use of deductive, logical reasoning ([13]). Westerners show a greater tendency to use a deductive approach in such situations. For example, Eastern students are more likely than Western students to reject logically valid arguments that contain implausible or unbelievable content ([99]). This cultural difference in reasoning is one of preference and not competence. The reasoning of East Asian students does not differ from that of Westerners when abstract content is used ([98]). Because of this cultural difference, there is the potential for East Asian students to encounter criticism when showing a preference for more inductive or intuitive approaches to problem solving ([131]).

Cultures also differ in the extent to which members of that culture display an uncritical acceptance of authority figures as reliable sources of information. For example, French students have been found to be less sensitive to the strength of expert testimony than Dutch students ([45]). Strength was defined in terms of the relation between an expert’s area of specialization and the subject about which the expert provided testimony. Expert testimony was considered strong when the expert testified about matters closely related to their area of expertise. As compared with Dutch students, the French students were less likely to distinguish between strong and weak expert testimony in judging whether or not to accept it. This cultural difference may be related to findings that obedience and respect for authority figures is more prevalent in primary/secondary education in France than in The Netherlands ([45]).

### 6.3. Gender

Gender roles can affect attempts to foster AR in the classroom. When instructions for engaging in deliberative argument emphasize the adversarial aspects of this kind of argument, boys, but not girls, view the activity in strictly competitive terms to the exclusion of key collaborative aspects of deliberative argument. Similarly, when instructions emphasize the collaborate aspects, girls, but not boys, tend to stress collaboration and cooperative to the exclusion of key adversarial elements ([5]). Thus, helping students appreciate the key balance between adversarial and collaborative elements that defines deliberative argument may require different framing for boys and girls ([8]). The collaborative elements may need greater emphasis with boys while the adversarial elements may need great emphasis with girls.

Even though female students have been found to construct more cogent argumentative essays than male students and to make more effective revisions in response to feedback ([9]; [100]), they sometimes display low confidence in their ability to write an effective essay ([107]). This mismatch between achievement and perceived competence is similar to that found for girls in regard to math and science ([154]). One factor that may contribute to this lower confidence is the argumentative style of boys in mixed gender groups. Male students are more likely to exhibit directness and boldness in making claims and justifying them, and they display greater comfort than female students with a disputative and competitive approach ([102]). For this reason, a peer group of mixed genders may not always support a female student’s perceived competence as a participant in deliberative arguments.

The replacement of face-to-face dialogue with computer mediated dialogue can reduce the prominence of interpersonal factors experienced by performance-oriented students, students from collectivist cultures, or female students in mixed gender groupings. Computer-mediated dialogue reduces the need to manage how one appears to others. As a result, students are more comfortable expressing viewpoints and critiquing the viewpoints of others. Arguers are less likely to reach consensus prematurely or to neglect criticisms of their position when engaging in dialogic argument via computer ([8]). Motivations such as a need to please or to be liked by one’s peers are less likely to prevail over the conviction that one’s claims are true and cogent. In addition, dialogue as text allows for reflection and review as well as reconsideration and revision.

## 7. Conclusions

The review presented in this article finds that many of the individual skills of AR are present in at least basic form early in development. This suggests that they do not need to be instilled so much as supported and refined ([88]). Perhaps the most significant milestone in the development of AR, however, is the coordinated and strategic use of these reasoning skills to conform to the goals and norms of deliberative argument. The difference between the presence of individual skills and their coordinated and strategic use is an important one ([106]; [116]). While evolution may have equipped children with precocious argument skills, these were selected for their effectiveness in establishing consensus. In the radically transformed information- and technology-rich human environment of today, however, consensus building cannot be the exclusive purpose of reasoning. Truth-finding needs to be given priority over consensus, and this requires an understanding of the nature of deliberative argument and the nature of knowledge, something which develops in adolescence given student exposure to educational settings that value dialogic argument ([80]; [59]; [93]). The findings from training studies indicate that students are best prepared to develop a more explicit understanding of deliberative argument and a more coordinated use of AR skills in early adolescence (middle school). Arguably, then, this should be the time when a classroom intervention would yield the most benefits. At the same time, there is good reason to expect significant success earlier, in pre-adolescence, if collaborative, dialogical activities are made an integral part of the classroom setting from early in children’s primary education.

The results of attempts at training AR skills and knowledge of deliberative argument are relatively clear with regard to the types of training methods that are most effective.

Training that is researcher- or teacher-led, featuring substantial direct instruction of AR skills within whole-class or large group settings, have demonstrated success in promoting AR skills and knowledge ([14]; [139]; [156]). However, these interventions have not always demonstrated the kinds of transfer effects across media and domains that would indicate the acquisition of generalized and self-regulated skills ([101]). By comparison, training in which the teacher serves as guide and facilitator to structured peer interaction have been more successful in this respect. The Argue With Me paradigm is one of the best interventions of this kind ([59]). Although even the more modest deployments of this paradigm involve an intensive immersion in dialogic argument that requires a significant commitment by staff and students, their effectiveness is clear.

Engagement of students with one another in regular, sustained dialogue-based activities, primarily within small groups, including dialogue with peers who espouse alternative viewpoints on a topic, is effective in promoting specific AR skills and strategies and an explicit understanding of the merits of deliberative argument. Preparation for these dialogues is critical and involves enabling students to acquire at least a rudimentary knowledge base in the topic. Preparation also includes small group discussion of reasons supporting each side and of potential counterarguments and rebuttals. The role of instructor, coach, or facilitator is to guide the preparatory activities, provide potential reasons that might be used to support a claim and provide feedback on reasons generated by the group members. A number of other training programs sharing several features with the AWM paradigm have also been successful (e.g., [96]; [121]; [123]).

The existing evidence suggests that effective efforts to promote AR should not limit their focus to AR skills themselves. Classroom interventions should also include reflective activities that help students recognize the relation between specific reasoning skills and the goals and norms of deliberative argument ([9]; [33]; [55]; [60]; [121], [122]). Only when students can think of the skills they are acquiring in terms of a meta-level understanding of the nature of deliberative argument can they utilize these skills strategically to achieve the goals of this type of dialogue. Reflective activities can also foster advances in epistemic cognition, i.e., advances in students’ understanding of the nature of knowledge and knowing. The development of epistemic cognition is itself linked to positive academic outcomes ([18]) and it is closely related to the development of AR ([80]; [54]; [155]). If students maintain an absolutist conception of argument wherein one side in a dispute fits the facts and is wholly right’ while the other side does not and is therefore ‘wrong’, there is little purpose or motivation to engage in argument. A process of fact-checking would be sufficient. Likewise, until valid knowledge is understood as resulting from standardized and transparent justification procedures, there is little point to argument since there is no basis for deciding between competing claims.

The use of computers as a medium for argumentative dialogue is common in successful training programs ([132]; [133]). Computer mediation represents an effective means for reducing the interpersonal concerns typically present in face-to-face argument. Students who are performance-oriented or whose cultural background underscores the risks of argument to the self and to the group are especially likely to benefit from conducting argument dialogues via computer. Computer-mediated dialogue helps students focus on the argument rather than the arguers and it supports student efforts to represent and reflect upon argument as both a process and a product.

Beyond the value of the medium itself, the use of scaffolding programs to support student engagement in computer-based argument has proven highly effective ([47]; [83]). There is now a burgeoning literature on the design and application of scaffolding programs for teaching AR skills as well as general knowledge of deliberative argument ([101]). One can expect that this form of technological support will expand further with the introduction of AI into the classroom. One of the most significant benefits of scaffolding is in reducing cognitive load. The information processing demands of participating in deliberative argument would seem to be substantial and extending the success of efforts to train AR into the primary school years is dependent upon how successful that training is in overcoming developmental and individual limitations in working memory capacity and cognitive control. And yet empirical research into the information processing demands of specific AR skills is lacking. Consequently, the design of computer scaffolding and other programming to reduce cognitive load relies on theory, computer simulation and modeling, and assessments of whether or not a given program promotes success. This is sufficient to proceed, but research into what kinds of processing demands are implicated in specific aspects of AR would help explain why certain training methods are successful or not.

The teaching of domain-general critical thinking abilities and AR skills in college has generally been successful ([50]), though the extent to which gains are maintained beyond college is not known. Given this success and given the benefits that can result from proficiency in deliberative argument, it is surprising that there has not been a greater investment in promoting critical thinking and AR in secondary or primary school. The research reviewed here strongly suggests that students can benefit from instruction in AR by middle school if not earlier. For students who are college-bound, beginning college with an appreciation of the value of deliberative argument would seem to be a clear asset. For those students who do not go to college, middle and secondary school represent the best, and perhaps the only, opportunity to develop the intellectual values and reasoning skills needed to participate in an information-rich, technological, and democratic society ([8]).

## Figures and Tables

**Table 1 behavsci-15-01700-t001:** Search Terms Used to Generate Candidate Papers for Inclusion in the Review.

Section 3: Academic Outcomes of AR	
**Concept A**	**Concept B**
AR	(Academic or School) Outcomes
Argument	Writing
Argumentation	Learning
Argument skills	Knowledge
Deliberative Argument	Argue (Arguing) to Learn
	Science, History, Math, Physics, Chemistry
	Academic Achievement, Critical Thinking
Section 4: Development of AR	
**Concept A**	**Concept B**
AR	Development
Argument	Child(ren), Adolescent
Argumentation	Childhood, Adolescence
Argument skills	Preschool, Early Childhood
Deliberative Argument	Middle School, High school
Section 5: Training of AR	
**Concept A**	**Concept B**
AR	Training
Argument	Teaching, Pedagogy, Instruction
Argumentation	Intervention
Argument skills	Motivation
Deliberative Argument	Culture
Learning to Argue	Gender

Note. For each of the above sections of the paper all searches involved the conjunction of Concept A terms and Concept B terms.

**Table 2 behavsci-15-01700-t002:** Key Elements of Argumentative Reasoning.

Claim	A statement one believes to be true.
Justification	Reasons or evidence for why the claim should be accepted
Counterargument (Challenge)	An attempt to refute a claim or its justification. The refutation could concern the truth, relevance, sufficiency, or other aspect of the claim/justification.
Rebuttal (Defense)	Response to a counterargument that defends the statement that has been challenged or that challenges the justification given for the counterargument
Concession	Rejecting one’s own statement following challenge.
Agreement	Accepting a statement by the opponent.
Example: Argument Concerning Diet	
TURN 1	
ANDREA	Eating meat is unhealthy. (Claim) Because it puts you at risk for obesity, high blood pressure, and hardening of the arteries. (Justification)
BRENDA	An appropriate amount of meat as part of a balanced diet does not put you at risk. (Counterargument)
TURN 2	
ANDREA	Any amount of meat is bad. Think about all the chemicals in meat—the preservatives, salt, antibiotics, dyes? These are unhealthy. (Rebuttal)
BRENDA	It’s not like vegetables don’t have these things in them. They spray fruits and vegetables with antibiotics and pesticides. (Counterargument)

## Data Availability

The author has given his consent to publish.

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
