# Peer review of "Argumentative Reasoning: Development, Training, and Relevance to Academic Outcomes"

_behavsci, 2025, doi:10.3390/bs15121700_

Round 1
Reviewer 1 Report
Comments and Suggestions for Authors
This is a very clear and well-written piece synthesizing research on argumentative reasoning. The topic is important, as the first part of the piece points out, and the author isn’t shy of raising contradicting claims or patterns in the research that don’t (yet) make sense.
I have two points of recommendation for the author. First, since this is a synthesis piece, it’d be useful to know how the author arrived at the set of studies reviewed – e.g., search procedures, how the author kept track of different findings, etc. Methods sections of articles in the Review of Educational Research can provide a helpful template; even if this journal does not require the level of detail specified in RER, providing some sense of the data corpus for the paper would be useful.
Second, this issue became especially acute for me in the section titled “Conclusion…”. There the author attempts to extrapolate from interventions into argumentative reasoning, but there’s no basis for how he came to his conclusions (e.g., “I categorized pedagogical methods and then graphed effect sizes in each category, inspecting….”). I also recommend making this conclusion section part of the body of the paper, as it is the result of the author’s synthetic analysis (similar to the other sections).
Finally, a note to the editor: by some byproduct of the reviewer selection process, I am not an expert in cognitive research on argumentative reasoning. So, other, more qualified reviewers’ comments about the substance of the paper should take precedence over mine.
Reviewer 2 Report
Comments and Suggestions for Authors
This manuscript offers an extensive, well-informed, and conceptually rich review of research on AR. The author demonstrates an impressive command of the literature and convincingly presents AR as a central cognitive and educational construct that connects reasoning, learning, and epistemic development. The article is logically organized and broadly comprehensive.
That said, the density of information and the occasionally uniform tone somewhat limit the paper’s readability and rhetorical impact. Some sections would benefit from more concise synthesis and clearer links between developmental evidence and educational practice.
Abstract
The abstract is informative and well-focused, effectively summarizing the paper’s scope, rationale, and significance. It could, however, more explicitly clarify how this synthesis advances or refines existing models of AR. A brief statement of the review’s distinctive contribution would strengthen its relevance for readers already familiar with the field.
Introduction
The introduction sets out a strong contextual motivation and situates the work within current debates on reasoning and democratic discourse. Yet it would benefit from a more explicit thesis or guiding question. At present, it reads more like a preface than an argument for the paper’s central claim. Establishing this conceptual through-line earlier would improve both coherence and focus.
Section 1
This section provides clear and detailed definitions of key components of AR. Nonetheless, it could be streamlined. Condensing some of the definitional material would free space for a more reflective discussion of why deliberative argumentation should take priority in education. A sharper articulation of AR’s broader significance, beyond its structural features, would make the section more compelling.
Section 2
The analysis of academic outcomes linked to AR is strong and well-documented. To improve readability and cohesion, consider adding brief integrative paragraphs after each subsection that draw together the main insights and emphasize their implications. These transitional syntheses would enhance flow and help readers follow the broader argument.
Section 3
The developmental account of AR is rich and well-grounded in empirical evidence. However, adding a short synthetic discussion at the end of this section, highlighting the general principles that emerge from the reviewed studies, would clarify what unites the findings. Explicitly identifying these takeaways would also create a smoother transition to the subsequent pedagogical implications.
Section 5
The final section is practical and insightful. To conclude on a more resonant note, consider ending with a concise, forward-looking paragraph that outlines broader implications for educational policy and teacher education. Such an ending would reinforce the manuscript’s social and civic significance.
Style, Coherence, and Scholarly Contribution
The manuscript maintains an appropriately formal academic tone. The prose is consistently precise, though at times slightly over-elaborate. Breaking up long paragraphs and varying sentence rhythm would make it more engaging.
Reviewer 3 Report
Comments and Suggestions for Authors
The paper presents a review of the link between argumentative reasoning and academic achievement among students in primary and secondary levels of schooling. The author could bring out the following in the discussion
a) What are the prerequisites at each level of schooling to facilitate better linkage?
b) Does cultural context affect the linkage, if so, how?
c) Are there any roles of gender and demographic characteristics that affect this linkage, and how children engage with AR? If so, what are the possible strategies to mitigate this? It would be good to explain how these characteristics affect (As mediating factors) the skills of reflection and exposure to dialogues and debates, thereby affecting overall AR skills.
